# Functional Interaction among K_Ca_ and TRP Channels for Cardiovascular Physiology: Modern Perspectives on Aging and Chronic Disease

**DOI:** 10.3390/ijms20061380

**Published:** 2019-03-19

**Authors:** Erik J. Behringer, Md A. Hakim

**Affiliations:** Department of Basic Sciences, 11041 Campus Street, Risley Hall, Loma Linda University, Loma Linda, CA 92350, USA; mhakim@llu.edu

**Keywords:** Ca^2+^-activated K^+^ channels, transient receptor potential channels, intracellular Ca^2+^ homeostasis, endothelium-derived hyperpolarization, cardiovascular disease, aging

## Abstract

Effective delivery of oxygen and essential nutrients to vital organs and tissues throughout the body requires adequate blood flow supplied through resistance vessels. The intimate relationship between intracellular calcium ([Ca^2+^]_i_) and regulation of membrane potential (V_m_) is indispensable for maintaining blood flow regulation. In particular, Ca^2+^-activated K^+^ (K_Ca_) channels were ascertained as transducers of elevated [Ca^2+^]_i_ signals into hyperpolarization of V_m_ as a pathway for decreasing vascular resistance, thereby enhancing blood flow. Recent evidence also supports the reverse role for K_Ca_ channels, in which they facilitate Ca^2+^ influx into the cell interior through open non-selective cation (e.g., transient receptor potential; TRP) channels in accord with robust electrical (hyperpolarization) and concentration (~20,000-fold) transmembrane gradients for Ca^2+^. Such an arrangement supports a feed-forward activation of V_m_ hyperpolarization while potentially boosting production of nitric oxide. Furthermore, in vascular types expressing TRP channels but deficient in functional K_Ca_ channels (e.g., collecting lymphatic endothelium), there are profound alterations such as downstream depolarizing ionic fluxes and the absence of dynamic hyperpolarizing events. Altogether, this review is a refined set of evidence-based perspectives focused on the role of the endothelial K_Ca_ and TRP channels throughout multiple experimental animal models and vascular types. We discuss the diverse interactions among K_Ca_ and TRP channels to integrate Ca^2+^, oxidative, and electrical signaling in the context of cardiovascular physiology and pathology. Building from a foundation of cellular biophysical data throughout a wide and diverse compilation of significant discoveries, a translational narrative is provided for readers toward the treatment and prevention of chronic, age-related cardiovascular disease.

## 1. Introduction

Endothelial cells lining the lumen of resistance arteries command changes in vascular diameter as needed to meet the metabolic demand of vital organs and tissues throughout the body. In particular, the relationship of intracellular calcium ([Ca^2+^]_i_) to the hyperpolarization of membrane potential (V_m_) is essential for a key signaling pathway underlying blood flow control, known as endothelium-derived hyperpolarization (EDH; i.e., activation of small- and intermediate-Ca^2+^-activated K^+^ (SK_Ca_/IK_Ca_) channels). In such a manner, endothelial cells coordinate with their surrounding smooth muscle cells via gap junctions for arterial relaxation and increased blood flow [1]. Physiological stimulation of EDH entails stimulation of G_q_-protein-coupled receptors (G_q_PCRs) and then an increase in [Ca^2+^]_i_ typically defined by initial Ca^2+^ release from the endoplasmic reticulum (ER) followed by a “plateau” phase of extracellular Ca^2+^ influx into the cellular interior through open transient receptor potential (TRP) channels [2,3]. Vascular TRP channels may be recruited by Orai for Ca^2+^ influx in response to cytosolic stromal interaction molecule (STIM) oligomers that form as a result of ER Ca^2+^ depletion [4,5,6,7]. The influx of Ca^2+^ contributes serves a dual role to (i) sustain activation of intracellular enzymes and plasma membrane ion channels for prolonged cellular function, and (ii) to refill ER Ca^2+^ (via smooth endoplasmic reticulum Ca^2+^ ATPase (SERCA) pump activity) to maintain consistent and repetitive physiological function. During elevation of overall [Ca^2+^]_i_, SK_Ca_/IK_Ca_ channels are activated, and the cell interior local to the inner leaflet of the plasma membrane increases in negative charge (“hyperpolarization”) due to K^+^ efflux through open SK_Ca_/IK_Ca_ channel pores [8,9]. Thus, the intimate relationship of endothelial [Ca^2+^]_i_ and V_m_ is integral to blood flow regulation and, therefore, indispensable for cardiovascular function.

The aim of this review is to resolve the interaction of K_Ca_ and TRP channel function with [Ca^2+^]_i_ and electrical signaling during physiology and pathology. The current working paradigm of the “microanatomy” of the [Ca^2+^]_i_-to-electrical signaling interface underlying EDH is located to endothelial projections that traverse through the internal elastic lamina to contact smooth muscle cells (Figure 1). Endothelial projections contain spatially proximal myoendothelial gap junctions, IK_Ca_ channels, TRP channels, and inositol 1,4,5-trisphosphate receptors (IP_3_Rs) of the endoplasmic reticulum [10,11,12]. When K_Ca_ channel function is present, robust V_m_ hyperpolarization occurs in response to sequential [Ca^2+^]_i_ release and Ca^2+^ influx, notably during treatment with a G_q_PCR agonist (e.g., acetylcholine (ACh) or adenosine triphosphate (ATP)) or hydrogen peroxide (H_2_O_2_), which mimics conditions of oxidative stress during old age. When K_Ca_ channel function is absent, an otherwise rapid V_m_ hyperpolarization response converts into slow V_m_ depolarization due to accumulation of intracellular positive charges in the form of Na^+^ and Ca^2+^ having entered through activated non-selective cation channels. With perspectives for future directions, we examine recent evidence demonstrating the functional contrasts among “normal” or “healthy” vascular aging and the development of chronic disease (e.g., hypertension, diabetes, heart failure, and coronary artery disease).

## 2. Significance of the Relationship of [Ca^2+^]_i_ and V_m_ in the Vascular Wall of Blood Vessels

Whether influenced by hormones, transmitters, or the shear forces of blood flow itself, the interplay between smooth muscle and endothelium of the vascular wall regulates resistance to blood flow in the form of vascular dilation or constriction [1]. Myoendothelial gap junctions between the two cell layers allow for the transmission of IP_3_, Ca^2+^, cyclic adenosine monophosphate (cAMP), cyclic guanosine monophosphate (cGMP), or the general distribution of electrogenic ions (e.g., Na^+^, K^+^, or Cl^−^) from endothelium to smooth muscle in response to stimulation of endothelial G_q_PCRs (e.g., muscarinic, purinergic) [13,14], or vice versa following smooth muscle α-adrenergic receptor stimulation [15]. Fundamentally, the central mediator underlying coordination of vascular reactivity is Ca^2+^, which activates endothelial nitric oxide synthase (eNOS) and the production of nitric oxide (NO) concomitant with stimulation of SK_Ca_/IK_Ca_ channels for EDH. The oxidoreductase activity of eNOS is described by the conversion of l-arginine to NO and l-citrulline with additional requirements of the Ca^2+^ sensor calmodulin, the co-factor tetrahydrobiopterin (BH_4_; composes the oxygenase domain with heme), and nicotinamide adenine dinucleotide phosphate (NADPH; reductase domain) [16]; deficiency in l-arginine or BH_4_ results in eNOS “uncoupling”, whereby reactive oxygen species are produced [17]. Smooth muscle relaxation to eNOS-derived NO occurs via protein kinase G (PKG)-dependent phosphorylation of multiple ion pumps and channels (e.g., SERCAs or voltage-gated K^+^ channels) in a manner that establishes a hyperpolarized smooth muscle V_m_, reduced smooth muscle [Ca^2+^]_i_ and, thereby, reduced myosin light-chain phosphorylation [18,19]. Calmodulin also plays a primary role in SK_Ca_/IK_Ca_ channel activation via the sensing of [Ca^2+^]_i_ using C- and N-lobe EF hand domains to open channel pores and allow K^+^ efflux from the endothelial cell. However, in studies using *Xenopus* oocytes and the inside-out patch clamp configuration to examine intracellular regulation of SK_Ca_ channels, it was found that the C-lobe may play a dispensable role for modulating Ca^2+^ affinity, whereas the N-lobe in particular constitutively stabilizes K_Ca_ subunits for activation [20]. The resulting hyperpolarization of endothelial V_m_ transmits to the smooth muscle via myoendothelial gap junctions [21,22], whereby L-type voltage-gated Ca^2+^ channels are deactivated, and in like fashion with the NO/cGMP/PKG pathway, smooth muscle [Ca^2+^]_i_ is ultimately reduced to promote vasodilation [23]. 

Original investigations of the structural resolution of myoendothelial gap junctions [24,25] and functional determinations of myography and electrophysiology [26] altogether revealed regional contributions of EDH vs. NO to vasodilation along the vascular network. In particular, myoendothelial gap junctions are composed of connexins (Cxns) Cx37, Cx40, and Cx43 [11,27,28] as required for the spread of EDH from the endothelium to the smooth muscle, a mechanism that plays a prominent role in small arteries and arterioles [29]. Shimokawa et al. showed that the contribution of EDH to endothelium-dependent relaxations rises as vessel size (diameter) decreases in six- to eight-month-old male rats [26]. In particular, the range of the contribution of EDH was >2-fold when extending from aorta (~30%) to the proximal (~46%) and then to the distal (~72%) mesenteric arteries, whereas trends in NO-dependent vasodilation were the opposite (aorta: ~56%, proximal: ~17%, distal: ~20%). It is also worth noting that the contribution of prostacyclin (PGI_2_) was negligible regardless of blood vessel size. Thus, when examining Ca^2+^ and electrical signaling underlying EDH or NO, it is important to consider the anatomical position of the arterial segment throughout the conduit and resistance blood vessel network feeding into each organ in the body. 

Altogether, regardless of source (intracellular release or plasma membrane entry), increased [Ca^2+^]_i_ plays a dichotomous role in the smooth muscle vs. endothelial cell layers (See Figure 1 Legend; smooth muscle [Ca^2+^]_i_ increase → depolarization → L-type Ca^2+^ channel activation → myosin light-chain phosphorylation → vasoconstriction vs. endothelial [Ca^2+^]_i_ increase → SK_Ca_/IK_Ca_ channel activation → hyperpolarization → myosin light-chain dephosphorylation → vasodilation) and maintains a narrow window of effective blood flow regulation [30,31] while preventing vascular rupture or ischemia. With some exception (e.g., direct PKG activation of myosin light-chain phosphatase and subsequent dephosphorylation of myosin light chain [32]), the cross-talk between [Ca^2+^]_i_ and V_m_ is the “master regulator” for the coordination of blood flow throughout vascular resistance networks regardless of the mode of the upstream cellular signaling pathway. The most direct bridge between these two physiological variables is EDH with SK_Ca_/IK_Ca_ channels as the transducers of increased [Ca^2+^]_i_ to hyperpolarization of the V_m_ throughout the vascular wall.

Recent perspective points to an initial rapid role for EDH during vasodilation following the onset of physical activity and skeletal muscle contraction, whereas NO signaling underlies a secondary prolonged but slower vasorelaxation for sustained blood flow per lumenal sheer stress [1]. It is also worth noting that the spatial domain of signaling for NO is on the order of hundreds of microns vs. thousands of microns for EDH along the vascular wall encompassing from large extraparenchymal arteries to capillaries. Furthermore, a phenomenon of G_q_PCR-stimulated “slow” Ca^2+^ waves (~100 µm/s vs. cm/s for electrical conduction) among and along the endothelial cell layer may govern the spatial activation of both NO and EDH [33,34]. Although, as described, Ca^2+^ waves occur within an order of timing most consistent with the production and signaling of NO. It is possible that the amplitude, distance, and/or speed of a Ca^2+^ wave can be enhanced by direct opening of SK_Ca_/IK_Ca_ channels to hyperpolarize V_m_, thereby increasing the electrical gradient for Ca^2+^ to move into cells through open TRP channels, a phenomenon known as “hyperpolarization-induced Ca^2+^ entry” [3]. In its entirety for intra- and intercellular signaling, this hypothesis was proposed >5 years ago [34], and now has support with findings of enhancement in NO production [35] and the feed-forward activation of EDH [3,36]. Of course, there are limits to the effectiveness of SK_Ca_/IK_Ca_ channel activation, vasodilation, and the delivery of blood flow, as we found that hyperpolarization of V_m_ >−60 mV results in current leak that reduces the spread of hyperpolarization among endothelial cells by more than half [37,38]. In addition to the diminishment of sufficient resting vascular tone [10], the consequence of the “over-activation” of SK_Ca_/IK_Ca_ channels entails significant charge loss through electrically “leaky” membranes, thereby impairing effective cell-to-cell vasoreactivity coordination of blood flow along vascular resistance networks [1,21]. Accordingly, as the relationship between V_m_ and the increase in [Ca^2+^]_i_ is only linear from ~−25 to −60 mV [3], and as only 10 to 15 mV of hyperpolarization from resting (~−30 to −40 mV) is needed for maximal vasodilation [39,40], “more” is certainly not “better” with respect to K_Ca_ channel function.

## 3. Significance of SK_Ca_/IK_Ca_ and TRP Channels in the Blood Vasculature

Properties of transmembrane ion channel activity include the electrical and driving forces on the ionic species that are permeant through the channel pore, channel conductance for ease of current flow, and the number of channels (*n*) multiplied by their individual probability of opening (P_O_). The SK_Ca_/IK_Ca_ channels are permeant to K^+^ ions which are favored for outward movement in accordance with the concentration gradient ([K^+^]_o_/[K^+^]_i_: ~0.030) despite an opposing electrical gradient (the intracellular side of the plasma membrane is negatively charged and attracts cations). The V_m_ must be as high as −90 mV at physiological temperature (37 °C) in order to bring transmembrane K^+^ flux to equilibrium (E_K_). For reference, vascular endothelial cells typically have a V_m_ of ~−30 to −40 mV during rest, and activation of SK_Ca_/IK_Ca_ channels alone using NS309 can increase V_m_ to E_K_ [37,41]. As a dominant form of K^+^ channel expressed in the endothelial membrane, the SK_Ca_/IK_Ca_ channels are tetrameric (four subunits), voltage-independent (from V_m_ ~−80 to +10 mV [37]), and are constitutively bound to calmodulin binding sites for [Ca^2+^]_i_ [42]. The physiological “agonist” is Ca^2+^, whereby the P_O_ of the channels is determined by the [Ca^2+^]_i_ needed to produce half-maximal activation for opening channel pores (K_0.5_) and cooperativity among the four subunits of the channel per number of Ca^2+^ ions (Hill coefficient; HC).

The SK_Ca_ channels were first identified for their role in the “after hyperpolarization” phase following the firing of individual action potentials in central neurons of the mammalian brain [43]. SK_Ca_ channels (K_Ca_2.1, K_Ca_2.2, and K_Ca_2.3, or SK1, SK2, and SK3), have a relatively “small” conductance (~5 to 20 pS), a high affinity for Ca^2+^ (K_0.5_: ~400 to 700 nM), and a steep dependence on Ca^2+^, whereby at least four Ca^2+^ ions (HC = 3.9 to 4.8) are involved in the cooperativity of channels subunits for gating the pore ([43]; SK messenger RNA (mRNA) extracted from human (SK1) and rat (SK2 and SK3) brain and cloned in *Xenopus* oocytes). Later, the K_Ca_2.3 (or SK3) channels in particular were demonstrated to play a role in endothelial cell regulation of vascular tone in mesenteric arteries and systemic blood pressure [44] using a SK3^T/T^ mouse model, whereby the K_Ca_2.3 gene promoter is governed by a tetracycline-sensitive transactivator protein [45]. Relative to wild type, the mesenteric arteries of SK3^T/T^ mice contained elevated expression of K_Ca_2.3 channels that were suppressed upon treatment with the tetracycline derivative doxycycline (DOX) to the extent that SK_Ca_ currents were ~25-fold lower in endothelial cells; thus, resting endothelial V_m_ was depolarized by ~14 mV. Also, vasoconstriction due to intravascular pressure (20 or 100 mmHg) or direct stimulation of smooth muscle α_1_-adrenergic receptors using phenylephrine was enhanced by >20% with blocking SK_Ca_ channels via apamin or removal of the endothelium in SK3^T/T^ mice (no DOX) or by genetically suppressing SK_Ca_ channel expression in SK3^T/T^ mice fed DOX. Furthermore, genetic deletion of SK_Ca_ channels increased diastolic (~94 to 118 mmHg) and systolic (~128 to 147 mmHg) blood pressure (mean arterial: ~105 to 128 mmHg) [44].

Relative to SK_Ca_ channels, IK_Ca_ (K_Ca_ 3.1, SK4, or IK1) channels have an “intermediate” conductance (=39 pS), reduced cooperativity of individual subunits per Ca^2+^ ion (>50%; HC = 1.7), and a similarly high Ca^2+^ affinity with a K_0.5_ of ~300 nM ([46]; IK1 mRNA extracted from human pancreas and cloned in *Xenopus* oocytes). The IK_Ca_ channels are generally absent in excitable cells such as neurons and cardiac myocytes [46,47,48] with an originally defined role characterized in immune cells [49]. However, similar to the role of SK_Ca_ channels for hyperpolarizing cellular V_m_, the importance of the IK_Ca_ channel is now well established for endothelium-dependent vascular function [50]. A comprehensive demonstration of the cardiovascular role of endothelial K_Ca_3.1 channels was achieved using a K_Ca_3.1^−/−^ (or IK^−/−^) mouse model with a genetic deletion of exon 4 coding the channel pore [51]. As a result, the overall endothelial K_Ca_ current densities were reduced by ~50%, while endothelial V_m_ hyperpolarization in response to ACh decreased by ~60%. Accordingly, measurements of maximal EDH-dependent vasodilation in response to ACh in isolated, pressurized carotid arteries ex vivo and cremaster arteries in vivo were less by ~30 to 50%. The physiological consequence entailed increased systolic and diastolic blood pressure by ~13 mmHg and ~12 mmHg, respectively (∆ mean arterial: ~14 mmHg) and mild left ventricular hypertrophy in K_Ca_3.1^−/−^ mice [51]. Thus, endothelial IK_Ca_ channels are fundamental for endothelial hyperpolarization underlying regulation of vascular tone, blood flow, and blood pressure, determinants that can altogether impact chronic cardiac remodeling as well. 

Following thorough characterization of individual genetic models for SK_Ca_ and IK_Ca_, further studies utilized SK3^T/T^IK^−/−^ offspring generated by interbreeding SK3^T/T^ and K_Ca_3.1^−/−^ mice, respectively [52]. This approach allowed for the delineation of the individual roles of SK_Ca_ and IK_Ca_ to EDH and regulation of blood pressure in a combinatorial manner. As expected, the combined genetic SK_Ca_ and IK_Ca_ deficiency eliminated endothelial K^+^ currents and substantially impaired (>50%) smooth muscle hyperpolarization of carotid arteries from IK1^−/−^/SK3^T/T^ mice (+DOX). Consequently, IK_Ca_ deficiency (IK^−/−^/SK^+/+^ mice) reduced ACh-induced EDH-mediated vasodilation by ~75%, whereas deficit of both channels (IK^−/−^/SK3^T/T^ + DOX mice) eliminated responses altogether (by ~99%). A reduction in ACh-induced vasodilation in either carotid arteries ex vivo or cremasteric arteries in vivo was not apparent in mice with a SK_Ca_ deficiency alone (IK^+/+^/SK^T/T^ + DOX). The mean arterial pressure relative to wild-type mice (~100 mmHg) modestly increased with SK_Ca_ deficiency (to ~106 mmHg), IK_Ca_ deficiency (to ~108 mmHg), or combined SK_Ca_/IK_Ca_ genetic deletion (to ~110 mmHg) [52]. An additional study of the double transgenic study concluded that SK_Ca_ channels, but not IK_Ca_ channels, played an integral role for vasodilation of cremasteric arterioles in response to tetanic muscle stimulation [53]. Also, it is also worth noting that the SK_Ca_ channel-driven vasodilation requires Cx40 endothelial and myoendothelial containing gap junctions [53], whereas IK_Ca_-dependent responses do not require Cx40 during hyperemia of the mouse cremaster tissue [54]. These general findings of the endothelial role of SK_Ca_ vs. IK_Ca_ channels were further reinforced using an endothelium-specific SK3 knock-out mouse model generated from crossing a floxed SK3 mouse with another that expresses endothelial Cre recombinase driven by the endothelial receptor-specific tyrosine kinase (or Tie2) promoter [55].

The majority of the work delineating the individual contributions of SK_Ca_ and IK_Ca_ channels to vasodilation and tissue hyperemia was performed using mesenteric (gut) arteries as an accessible and abundant source of resistance arteries in the body [56,57,58]. However, to a lesser extent, studies also resolved clear contributions of respective K_Ca_ channel subtypes in the microcirculation of the eye [59], brain [60,61,62], heart [63,64], lung [65,66], skeletal muscle [8,67], and kidney [68,69,70]. While it is clear that both SK_Ca_ and IK_Ca_ channels are integral to overall cardiovascular health (systemic vascular resistance and blood pressure control), respective K_Ca_ contributions underlying hyperemia throughout organs may vary. A common feature at the biophysical level in isolated endothelium is that IK_Ca_ (vs. SK_Ca_) channels play a predominant role in the regulation of V_m_ and membrane resistance during steady-state and pharmacological conditions in the presence of direct (e.g., NS309 and SKA-31) or indirect (e.g., ACh and ATP) K_Ca_ openers [8,71]. At the level of vasoreactivity and regulation of blood flow, evidence supports SK_Ca_ channels as the “affinity” component (sensitivity of vasodilation/hyperemia per unit stimulus), whereas IK_Ca_ channels define “efficacy” (amplitude of vasodilation/hyperemia per unit stimulus) [51,52,72].

Equipped with an enhanced electrical gradient via hyperpolarization [3] in accordance with a ~20,000-fold transmembrane concentration gradient [73], Ca^2+^ influx through open Ca^2+^-permeant TRP channels may serve at least two purposes: (i) to sustain the duration of elevated [Ca^2+^]_i_ for prolonged control of blood flow, and (ii) to facilitate Ca^2+^ refilling of the endoplasmic reticulum to maintain continued, repetitive vascular function [8]. Note that endothelial TRP channels involved in EDH can be homo- or heteromeric among families and respective isoforms: canonical (TRPC; isoforms 1, 3, 4, 5, and 6), vanilloid (TRPV; isoforms 1, 3, and 4), ankyrin (TRPA; isoform 1), and polycystin (TRPP; isoform 2) [74,75]. Functional examples of heteromeric configurations of vascular TRPs include TRPC3-C4 [76], TRPV4-C1 [77,78,79], and TRPV4-C1-P2 [80]. Due to recent breakthroughs in pharmacological and genetic tools [81,82], the role of TRPV4-containing channels is the most widely studied thus far [3,83,84].

In general, most experimental evidence demonstrates that TRP channel function produces EDH via [Ca^2+^]_i_ activation of SK_Ca_ and IK_Ca_ channels governing vascular dilation, thereby promoting increases in blood flow (Figure 2A). However, TRP channels can regulate EDH in a positive or negative manner based on relatively permeability of Ca^2+^ to Na^+^ ions of a particular TRP channel configuration. Whereas TRPC1-containing channels are relatively low conductance (~16 pS) and non-selective (P_Ca_/P_Na_ ~1:1; [85]), TRPC6 (~35 pS, P_Ca_/P_Na_ ~5:1; [86]) or TRPV4 (~90 pS, P_Ca_/P_Na_ ~6:1; [87]) channels are examples of high conductance, dominant for mediating Ca^2+^ (vs. Na^+^) influx. Consequently, TRPC1-containing channels in mouse resistance arteries mediate a net depolarizing influence on V_m_, counteracting EDH, whereas TRPC6- [88] and TRPV4-containing channels [83] augment EDH in the presence of functional SK_Ca_/IK_Ca_ channels. Depolarization of endothelial V_m_ is due to relatively higher Na^+^ permeability and influx, whereas net hyperpolarization is a result of prominent influx of Ca^2+^ because, unlike highly electrogenic Na^+^ ions, Ca^2+^ ions are primarily second messengers that rapidly bind with intracellular proteins such as calmodulin to activate SK_Ca_/IK_Ca_ channels (as one example). Similar to endothelial TRPC1, smooth muscle TRP melastatin (TRPM, isoform 4; ~25pS, not Ca^2+^ permeant [89]) channels also permit Na^+^ influx to depolarize smooth muscle V_m_, thereby activating L-type voltage Ca^2+^ channels for myogenic constriction [90]. Remarkably, blocking TRPM4 channels in rat mesenteric arteries using 9-phenanthrol also hyperpolarizes endothelial V_m_ via apparent enhancement of IK_Ca_ channel function [91]. Also, in mouse collecting lymphatic endothelium, where TRPV4 channels are present but functional SK_Ca_/IK_Ca_ channels are absent (i.e., no EDH), depolarization of V_m_ in response to TRPV4 channel activation occurs almost exclusively due to Na^+^ influx [92] (Figure 2B). Subtle variations of positive or negative regulation of EDH will depend on oligomeric configurations of the TRP channel to govern permeability of one form of cationic species vs. another, and, in this regard, much remains to be resolved in physiological study models vs. heterologous cell culture systems. 

Since original myoendothelial investigations around the turn of the 21st century [24,25], the use of transgenic animal models and confocal microscopy greatly advanced cardiovascular studies of “microdomain” signaling among spatially localized K_Ca_ and TRP channels. The Taylor laboratory group in particular developed auto-detection and analysis algorithms to quantitate discrete Ca^2+^ events throughout cellular interior with primary data outputs of ∆[Ca^2+^]_i_ event amplitude, frequency, duration, and spatial spread [93,94,95]. Throughout such studies, mesenteric arterial endothelium was examined using an en face study model, whereby experimentally feasible arteries (diameter: ~300 µm) for this protocol were cut open longitudinally and pinned down into silicone with the endothelium facing up toward the microscope objective. In comparison to wild type, the number of ACh-induced Ca^2+^ events during “plateau” responses (predominant TRP channel activity) are reduced by ≥60% in genetic elimination of IK1 (IK1^−/−^), whereas basal Ca^2+^ dynamics are similar [95]. In tandem with elimination of IK1, genetic SK3 suppression (IK1^−/−^/SK3^T/T^ + DOX) or SK3 overexpression (IK1^−/−^/SK3^T/T^ − DOX) has minimal effects on the occurrence of Ca^2+^ events under both basal and ACh-stimulated conditions [95]. In addition to removal of extracellular Ca^2+^, pharmacological corroboration of these findings was demonstrated with a block of SK_Ca_ and IK_Ca_ channels (apamin + charybdotoxin) or TRPV4 channel block with HC-067047, whereby ACh Ca^2+^ dynamics in wild-type arteries were reduced to the level of mice genetically deficient for both SK3 and IK1 (IK1^−/−^/SK3^T/T^ + DOX) [95]. A follow-up study characterized endothelial dynamic Ca^2+^ signals during basal and conditions of Substance P-stimulated vasorelaxation in swine coronary endothelial cells. This investigation detected endothelial spatial and temporal events governing arterial tone that were not apparent using averaged, whole-field measurements of [Ca^2+^]_i_ [96]. With the use of an endothelium-specific SK3 knock-out mouse model, it was found that the coupling of SK3 and TRPV4 channels in mesenteric arteries appears to elicit large-amplitude and slow-decay Ca^2+^ kinetics that may be consistent with relatively long-term physiological delivery of blood flow in response to metabolic demand of active tissues [55]. 

Another foundational study demonstrated how small (diameter: ~100 µm) mesenteric endothelial TRPV4 channel-mediated Ca^2+^ “sparklets” activate SK_Ca_ and IK_Ca_ channels using wild-type, endothelial genetic Ca^2+^ sensor (GCaMP2Cx40) and TRPV4^−/−^ mice and customized Ca^2+^ event detection software; only a few open TRPV4 channels are required for maximal vasodilation [83]. Later, it was found that this coupling between endothelial K_Ca_ and TRP channels was due, at least in part, to the presence of protein kinase C (PKC) and PKC-anchoring protein AKAP150 or, otherwise, hypertension would ensue [12]. However, this scenario can change depending on vessel type as, in mouse pulmonary arteries, TRPV4 Ca^2+^ “sparklets” preferentially activate eNOS in a negative feedback manner, whereby NO-activated PKG inhibits cooperative opening of TRPV4 channels [66] (Figure 2A). Remarkably, a decrease in intralumenal pressure on its own (from 80 mmHg to ≤50 mmHg of rat cremasteric arterioles) can significantly increase TRPV4-mediated Ca^2+^ events that selectively activate IK_Ca_ vs. SK_Ca_ channels, leading to a loss of myogenic tone and blood flow autoregulation [10]. Also, using a novel gradient index (or GRIN) fluorescence microendoscopy method positioned inside of rat carotid arteries, ACh-induced widefield Ca^2+^ events (summation of IP_3_R release and TRP Ca^2+^ influx) decreased with increasing pressure (60, 110, and 160 mmHg), an effect that was diminished during late middle age (18 mo) vs. youth (3 mo) [97].

Although studied to a lesser extent vs. TRPV4 channels, the activation of SK_Ca_ and IK_Ca_ channels occurs downstream of other novel families and/or isoforms of TRP channels as well. In particular, TRPV3 channels contain ~2-fold the unitary conductance and Ca^2+^ permeability (~150 to 200 pS, P_Ca_/P_Na_ ~12:1; [98]) of TRPV4 channels [87] and, thus, may be more pertinent for activating eNOS and/or EDH. With use of total internal fluorescence (TIRF) microscopy and the oregano monoterpenoid carvacrol (TRPV3 activator; [99]), TRPV3-to-SK_Ca_/IK_Ca_ channel coupling was characterized in rat cerebral parenchymal arterioles [100]. Notably, carvacrol-induced vasodilation was not sensitive to block of NO and cyclooxygenase signaling and was almost completely abolished upon conditions of either SK_Ca_ or IK_Ca_ block alone [100]. Another TRP channel of the ankyrin family (TRPA1; activated by mustard oil, P_Ca_/P_Na_ ~1:1; [101]) was found to contribute to endothelial SK_Ca_ and IK_Ca_ channel activation as well and in a manner insensitive to NO and cyclooxygenase signaling [102]. Despite very modest Ca^2+^-permeant properties vs. TRPV3- and TRPV4-containing channels, TRPA1 channel function may be involved in microdomain myoendothelial signaling for activation of K_Ca_ channels during physiology [103] and conditions of enhanced oxidative signaling [104]. Finally, in similar fashion to TRPV3, TRPV4, and TRPA1, the role of TRPC3 (~70 pS, P_Ca_/P_Na_ ~1.5:1; [105]) was thoroughly characterized for its coupling to SK_Ca_/IK_Ca_ channels in mouse cerebral arteries [71] and rat mesenteric arteries [106] using a selective antagonist as the pyrazole compound Pyr3. 

When interpreting studies on TRP channel expression and function, it is important to bear in mind that TRP channels are tetrameric in their physiological form and may contain only one isoform (homotetrameric) or two to four different isoforms (heterotetrameric). Parameters of conductance and Ca^2+^ permeability are primarily determined in heterologous cell culture systems that only express homotetrameric TRP channels to determine parameters of individual TRP isoforms. Furthermore, genetic tools (e.g., TRP knock-out animals) do not necessarily manifest results consistent with pharmacological agents (e.g., TRP blockers) [3,107], as the former method prevents genetic expression of the TRP isoform and perhaps decreases the total number of functional TRP channels overall, whereas the latter strategy modulates a post-translational TRP channel product. Again, for pharmacological interventions, heterotetrameric TRP channel configurations were considered and, thus, potential insensitivity to agonists or antagonists were demonstrated to be effective for homotetrameric channels formed in a heterologous system. 

## 4. Impact of Adrenergic Tone on SK_Ca_/IK_Ca_ and TRP Channels & Significance of In Vivo vs. Ex Vivo Observations

A central (and extremely complex) research topic entails how the stimulation of smooth muscle α-adrenergic receptors (α1ARs) influences the interaction between endothelial K_Ca_ and TRP channels during “myoendothelial feedback” [24,30,31]. In isolated mouse mesenteric arteries, activation of smooth muscle α1ARs elicits vasoconstriction for ~2 min followed by a slow dilation induced by increases in endothelial TRPV4-mediated Ca^2+^ influx and activation of K_Ca_ channels, a mechanism that may cease prolonged vascular resistance and hypertension as observed in TRPV4^−/−^ mice [108]. Whether the ionic species directly passing from smooth muscle to endothelial cells through gap junctions is in the form of IP_3_ (activator of IP_3_Rs and Ca^2+^ release from endothelial ER) [109], Ca^2+^ [14], or perhaps both [110], in the context of animal species (e.g., rat vs. mouse), blood vessel type (e.g., mesenteric vs. skeletal muscle) and/or experimental conditions (e.g., ex vivo vs. in vivo) remains a topic of active investigation. In addition, there is a complex interplay among the relative contributions of endothelial NO, SK_Ca_ channels, and IK_Ca_ channels as it pertains to the coordination of local and conducted vasodilation signaling. As a key example, a recent study demonstrated that isolated rat mesenteric arteries in the presence of norepinephrine mediate myoendothelial feedback primarily through activation of IK_Ca_ channels, whereas, with ongoing physiological sheer stress to blood flow, feedback is in the form of SK_Ca_ channel activation and NO production in the intact mesenteric network [111].

Despite efforts for experimental controls and proposals for simplified working models across vascular types and methods of pre-constricted tone, studies of myoendothelial feedback remain empirically convoluted by nature. To begin with, it is inherently difficult to delineate Ca^2+^ signaling events among respective smooth muscle and endothelial cell layers with common molecular targets (e.g., TRP channels and ER IP_3_Rs) of interest. Next, there are factors of interventional drug delivery (ablumenal vs. lumenal), permeability (cell-permeant vs. extracellular), and target selectivity to contend with. In addition, whether via means of a chemical fluorescent dye or genetic sensor, compaction or dilution of Ca^2+^ sensor molecules can alter with cellular mechanical movement regardless of distinct stimuli or pharmacological interventions. Also, as in the case of in vivo experiments, halogenated (e.g., isoflurane) or injectable (e.g., pentobarbital) anesthetics can influence vascular function on their own toward the alteration of endothelial function [112,113]. Furthermore, even agents that are commonly used for cell/tissue-specific genetic manipulations (e.g., tamoxifen (estrogen receptor partial agonist/antagonist) and DOX (tetracycline antibiotic)) and, interestingly, the choice of vehicle solvent (e.g., sunflower vs. peanut oil) [114] may introduce experimental artefacts. Thus, scientific corroboration among complementary methods such as a vascular co-culture method [115], pressure myography [116], microendoscopy imaging [117], and intravital microscopy [118] helps to clarify myoendothelial signaling events during health and disease.

## 5. Role of Familial Mutations in K_Ca_ and TRP Channels in the Emergence of Cardiovascular Disease

The cardiovascular consequences of deliberate experimental mutations of major components of EDH (K_Ca_ and TRP channels, and gap junction Cxns) were discussed, but genetic mutations are also present during inheritable diseases. A single autosomal dominant mutation in *KCNN3* (guanine (mutant) substitution for cytosine (wild type) at nucleotide 1348 in coding DNA (or c.1348 C>G) and the K_Ca_2.3 channel subunit (leucine (mutant) substitution for valine (wild type) at amino acid 450 (or V450L)) underlies development of idiopathic non-cirrhotic portal hypertension [119]. Also, a variety of single-nucleotide polymorphisms for *KCNN3* (chromosome 1q21 locus) are involved in lone atrial fibrillation [120], presumably due to dysregulation in the repolarization of action potentials in cardiac atria. As cardiac myocytes are also rich in gap junctions for coordinating electrical activity, it is not surprising that germline heterozygous missense mutations in *GJA5* (Cx40; e.g., phenylalanine for isoleucine, I75F [121]) also underlie lone atrial fibrillation. Autosomal dominant *KCNN4* (K_Ca_3.1; “Gardos” channel) mutations on chromosome 19 lead to decreases in erythrocyte production ([122]; Diamond–Blackfan anemia) and health ([123]; xerocytosis, dehydration due to excess K^+^ and H_2_O loss). TRP “channelopathy” mutations are linked to an assortment of disorders throughout the entire body, including cardiovascular maladies of ischemic heart disease, hypertension, and atherosclerosis (see review [124]). However, there are a few adequately described cases of familial mutations accompanying cardiovascular pathology, including *TRPC3* (Williams–Beuren syndrome, deletion of 26 to 28 genes in 7q11.23; blood vessel stenosis [125]) and *TRPM4* (progressive familial heart block type I, missense mutations in 19q13.32, and gain of function; atrioventricular block [126]). 

Although not discussed here, inheritable mutations in K_Ca_ channels, TRP channels, and Cxns also involve prominent neurological (e.g., Parkinson’s) and developmental (e.g., skeletal dysplasia) disorders. Further investigation is needed to delineate de novo germline mutations from post-zygotic mutations that arise and provide susceptibility to development of cardiovascular pathology (e.g., arrhythmias or hypertension) with advancing age and select conditions of environmental exposure. In addition, albeit indirectly, the functional efficiency of EDH may also be affected by germline and somatic mutations of the upstream (e.g., *CHRM3*; muscarinic type 3 G_q_PCR, M_3_R [127]) and/or downstream (e.g., *MYLK*; myosin light-chain kinase, MLCK [128]) signaling components of vascular function. As of the present, human genetic mutations in key components involved in EDH appear extremely rare while not selective toward cardiovascular impairments alone. As a clearer foundation for understanding mammalian cardiovascular function and pathology, focus on the impact of aging and associated pathology (primarily using rodent models) is discussed next. 

## 6. Endothelial SK_Ca_ and IK_Ca_ during Aging and Chronic Pathology

The leading cause of death in the United States is cardiovascular disease [129] with aging [130,131] and age-related vascular endothelial dysfunction [132,133] as key risk factors. A central mechanism of endothelial function for vasodilation and coordination of blood flow along and among vascular networks is EDH [8,134]. The physiological consequences of enhancement or diminishment of EDH during development of disease relative to young, healthy cardiovascular conditions remain altogether unclear. It is worth noting that IK_Ca_ function was found to increase in coronary arteries [135] and saphenous arteries [136] of obese rats, and mesenteric arteries of hypertensive [137] and diabetic [138] rats. At least in part, EDH compensates for reduced NO bioavailability during diabetic conditions in coronary arteries of dogs [139] and subcutaneous arteries of humans [140]. In aortae of apolipoprotein E knock-out mice, basal K_Ca_ channel function appears to be reduced with enhancement of peak hyperpolarization to indirect (ACh) and direct (SKA-31 or NS1619) stimulation of the channels [141]. Remarkably, EDH may also emerge as a prominent contributor to the dilation of cremaster arterioles in developing mice of 15 weeks of age fed a high-fat diet continuously, beginning from at least four weeks prior to conception vs. control or temporally split control/high-fat diet treatments [142]. With respect to aging, increased enhanced IK_Ca_ function was observed in the aortae of 75- to 100-week-old mice [143] and skeletal muscle feed arteries of 24- to 26-month-old mice [38,144]. Remarkably, large-conductance Ca^2+^-activated K^+^ (BK_Ca_) channels also emerge in vascular endothelium as a result of conditions associated with chronic hypoxia (48 h), forming microdomain arrangements among Caveolin-1, TRPV4 channels, and BK_Ca_ channels (rat gracilis arteries) [145]. It is worth noting that, either on its own or in combination with deficient gap junction transmission, K_Ca_ over-activation may significantly diminish conducted hyperpolarization and/or vasodilation during old age (≥24-month-old mouse skeletal muscle arteries [38], and ≥64-year-old human coronary arteries; [63]), hypertension (Cx40-deficient mice [146]), hyperhomocysteinemia (cystathionine β-synthase mouse gluteus maximus arterioles [147]), and chronic hyperglycemia (diabetic mouse mesenteric arteries [148]). Currently, the upregulation of EDH during aging and development of chronic pathology is best explained by enhanced oxidative signaling (Figure 3). In brief, superoxide (O_2_^•−^) production emerges from a variety of intracellular sources (NADPH oxidases, mitochondria, uncoupled NO synthase, and xanthine oxidase) which inactivates NO to peroxynitrite (ONOO^−^) [149,150] and, via superoxide dismutase, O_2_^•−^ is rapidly converted to H_2_O_2_ with spontaneous breakdown products such as hydroxyl radicals (OH^•^) [151] which increase EDH [8,38,152]. Thus, with this mechanistic basis in mind, SK_Ca_/IK_Ca_ channel function may “compensate” for sustaining local vasodilation [153] at the expense of a restricted spatial domain of the spread of coordinated blood flow due to “leaky” endothelial plasma membranes [8]. 

On the flip side, there is support that the contribution of EDH may also decrease during conditions of cardiovascular disease. In rats, respective contributions of NO and EDH to the dilation of mesenteric arteries are reduced during diabetes [154] or of kidney arterioles at 18 months of age relative to 3 months [155]. It was also determined that physiological contribution of SK_Ca_/IK_Ca_ channel function of saphenous arteries was less in 34- and 64-week-old (vs. 12-week-old) mice [156]. Finally, recent reports also conclude that EDH-dependent dilation of mesenteric arteries decreases in genetically hypertensive rats [157] or in 32-week-old rats fed a high-fat and/or high-fructose diet beginning at four weeks of age [158]. Discrepancies in conclusions for remodeled EDH function can be attributed to several differences among studies (e.g., diet application protocol or vascular study model). However, with consideration for “healthy” aging alone, perhaps impaired vascular IK_Ca_ channel function is not a consistent observation for rodents >20 months of age.

Recent support of SK_Ca_/IK_Ca_ and TRP channel function as a “double-edged sword” that promotes the gain or loss of blood flow control is intriguing and important for a modern understanding of acute and chronic cardiovascular disease. The extent of SK_Ca_/IK_Ca_ channel activation coincident with V_m_ ≥−60 mV may be excessive [3,38] as it pertains to the loss of myogenic tone and significant impairment of blood flow control [10]. Remarkably, pharmacological block or genetic deletion of IK_Ca_ channels may actually serve as protection as demonstrated in K_Ca_3.1^−/−^ mice that avoid an otherwise fatal pulmonary circulatory collapse in response to the activation of Ca^2+^-permeant TRPV4 channels with GSK1016790A (high dose: 10 nM [159]; see Reference [160] for a thorough review on the interaction of pulmonary K_Ca_ and TRPV4 channels). Furthermore, direct genetic or pharmacological inhibition of TRPV4 prevents inflammatory cytokine signaling and endothelial dysfunction in septic mice (via cecal ligation and puncture or injection of tumor necrosis factor α or lipopolysaccharide) [161] and cardiac left ventricular cell/tissue damage in aged mice (24 to 27 months) exposed to hypoosmotic conditions (250 mOsm vs. physiological, ~300 mOsm) representative of ischemia–reperfusion injury [162]. Theoretically, enhanced smooth muscle α-adrenergic receptor activation via perivascular sympathetic nerves during aging and chronic vascular disorders (e.g., hypertension) may further overstimulate endothelial IK_Ca_ channels in response to high IP_3_ and Ca^2+^ concentrations transmitted from smooth muscle through myoendothelial gap junctions [1,21]. Thus, in conditions of excessive stimulation of G_q_PCRs, such as adrenergic receptors or TRPV4 channels that ultimately cause supraphysiological increases in vascular wall [Ca^2+^]_i_, a deleted or diminished role for endothelial IK_Ca_ channels may actually alleviate the consequences of pathological burden of [Ca^2+^]_i_ associated with the development of acute and chronic disease.

## 7. What May Be Next for Investigative Studies of Endothelial Function: Novel Physiological and Pharmacological Molecular Signaling Pathways

The number of “endothelium” publications recognized by the National Library of Medicine significantly increased from ~5000 in the early 1990s [163] to the present with >150,000. Under this heading, studies of EDH as they relate to the interaction of K_Ca_ and TRP channels are a rapidly growing research landscape for resolving cardiovascular physiology (rest and exercise) and the development of chronic disease. In general, characterizations of the essential components of EDH (G_q_PCRs, IP_3_Rs, K_Ca_, and TRP channels) were thoroughly investigated across animal species and blood vessel types as discussed in this review. Finally, as a key intracellular organelle for the integration of Ca^2+^ and oxidative signaling in the vasculature, a physiological role for endothelial mitochondria is being investigated during ATP-dependent control of Ca^2+^ homeostasis [164], enhanced EDH during vascular aging [144], and metabolic stress during conditions of hyperglycemia [165] as a few recent examples. Also, additional clarification regarding endothelial mitochondrial K_ATP_ channels [166] and BK_Ca_ channels [167], and delineation of their actions of depolarization of the inner mitochondrial membrane vs. the well-characterized plasma membrane K^+^ channels (hyperpolarization of the plasma membrane) is required. 

In addition to NO, there are other gases now recognized to play a role during blood flow control including hydrogen sulfide (H_2_S) and carbon monoxide (CO). As a reducing agent (vs. oxidizing H_2_O_2_), H_2_S is generated from substrates homocysteine (via cystathionine β-synthase), cysteine (3-mercaptopyruvate sulfurtransferase), and thiosulfate (cystathionine γ-lyase), [168]. Overall evidence supports H_2_S as a vasodilator that increases TRPV4 channel-dependent Ca^2+^ events and activation of endothelial BK_Ca_ channels of rat mesenteric arteries [169]. H_2_S generated by cystathionine γ-lyase in particular may activate endothelial SK_Ca_ and IK_Ca_ channels for vasodilation of mouse mesenteric arteries [170]. CO is produced from heme via the actions of heme oxygenase [171]. Although there is some evidence to suggest that CO is integral to endothelial function [167] and vasodilation [172], further investigation is needed.

There are several other endogenous and therapeutic molecules (lipids and polyphenols) that operate via endothelium-dependent vasodilation. In particular, legalization and social acceptance for the recreational and medicinal use of cannabis among American populations increased since 2007 [173]. Thus, the investigation of cannabinoids is perhaps currently among the highest of priorities in biomedical research. Although reports are presently scarce, there is evidence to support that the endocannabinoid 2-arachidonoylglycerol induces vasorelaxation of rat mesenteric arteries as dependent on K_Ca_ and TRPV channels [174]. In human mesenteric arteries, this vasodilation appears to be dependent on cyclooxygenase metabolism and prostanoid signaling [175]. Recent evidence also suggests that endothelial cannabinoid receptors are not existent, and vasodilatory effects are directly mediated via BK_Ca_ channels [176]. As representative of studies testing cannabis Δ^9^-tetrahydrocannabinol (THC; CB1 receptor agonist) on vasoreactivity, there is evidence demonstrating the opposite as well, whereby the application of THC reduces EDH-dependent vasorelaxation in mouse arteries [177]. Regardless, a consensus is presently lacking across the mechanisms of action of various cannabinoids, respective pharmacological kinetics, presence/classification of vascular cannabinoid receptors, and contribution of K_Ca_ channels vs. NO vs. cyclooxygenases among animal species and blood vessel types. Other studies support the role of endothelial K_Ca_ channel function during arterial treatment with omega-3 polyunsaturated acids [178], fruit-derived polyphenols [179], *Rhododendron* flavonoids [180], or sex hormones (testosterone [181] and estrogen [182]). All of such studies and therapeutic strategies that are either ancillary to, or perhaps circumvent, classical G_q_PCR pathways (e.g., purinergic and muscarinic) are in need of further investigation.

## 8. Summary and Conclusions

Cardiovascular disease is the number one killer of Americans [129] with age-related vascular endothelial dysfunction at the center [132,133]. Coordinated blood flow and the perfusion of organs depend on the endothelial lining of major arteries and respective microcirculatory networks (arterioles and capillaries) that travel ≥100 km throughout the body [183,184]. The signaling pathway integral to the promotion of blood flow is endothelium-derived hyperpolarization (EDH). EDH sequentially entails G_q_-protein-coupled receptor stimulation in intracellular Ca^2+^ release from the endoplasmic reticulum and Ca^2+^ influx through transient receptor potential (TRP) channels to activate Ca^2+^-activated K^+^ (K_Ca_) channels and produce hyperpolarization [8]. The structural and functional arrangement among TRP and K_Ca_ channels in the blood vasculature is diverse across animal species, vessel type, and modes of local vs. conducted signaling to tune blood flow. Under- or over-activation of EDH is associated with aging and/or the development of chronic cardiovascular disease.

The future of cardiovascular physiology will likely entail progressive recruitment of state-of-the-art methods in the form of genetics, pharmacology, and engineering (e.g., stem-cell therapy [185], nanoparticle delivery of antioxidant molecules [186], and “blood–brain barrier on a chip” [187]) to examine classical components of genetics, structure, and function using cell/tissue/animal models of aging and disease. However, a present priority includes comprehensively defining a role of endothelial mitochondria as a nexus for vascular Ca^2+^ and oxidative signaling and modulatory input of novel molecules in the form of gases, lipids, hormones, and phytochemicals. Regardless of physiological examination and therapeutic strategies, we anticipate that fully understanding mechanisms underlying blood flow regulation as it relates to the aging cardiovascular system will ultimately lead to the prevention of chronic disease.

## Figures and Tables

**Figure 1 ijms-20-01380-f001:**
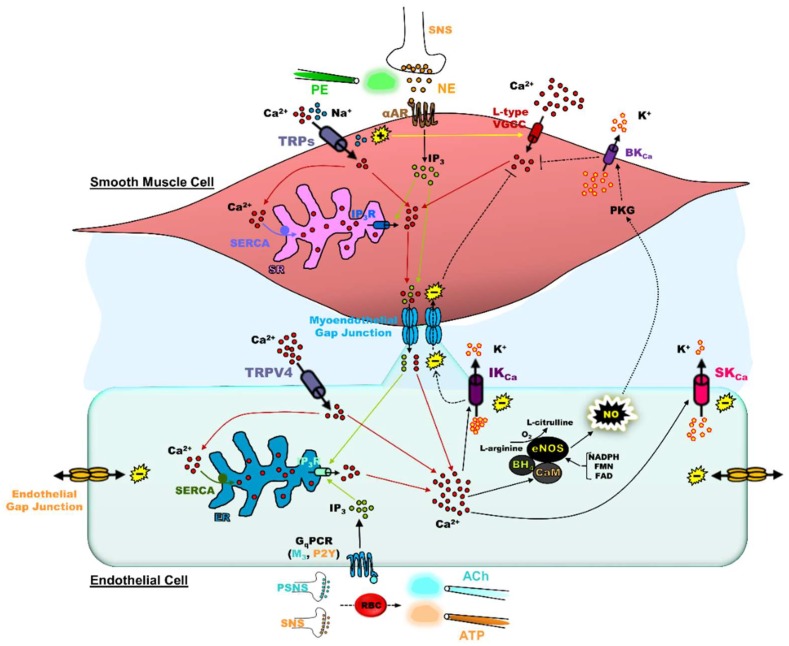
Anatomy of endothelium-derived hyperpolarization and myoendothelial coupling. Endothelium: Physiological stimulation of endothelial G_q_-protein-coupled receptors (G_q_PCRs; muscarinic (M_3_), purinergic (P2Y)) occurs through neurotransmitter secretion (parasympathetic nervous system (PSNS), acetylcholine (ACh); sympathetic nervous system (SNS), adenosine triphosphate (ATP)) or ATP release from red blood cells (RBCs) as primary examples. ACh and ATP are pharmacologically applied using bulk flow or focal delivery via pipettes (iontophoresis and pressure ejection). Following endothelial G_q_PCR activation, inositol 1,4,5-trisphosphate (IP_3_) is produced, which in turn activates IP_3_ receptors (IP_3_Rs) to release Ca^2+^ from the endoplasmic reticulum (ER) into the cytosol. ER Ca^2+^ release is observed as the initial “peak” response in ∆[Ca^2+^]_i_. The ER Ca^2+^ stores are filled or refilled through uptake of Ca^2+^ from the cytosol into the ER through smooth endoplasmic reticulum Ca^2+^ ATPase (SERCA) pumps to sustain repetitive physiological signaling. The influx of Ca^2+^ occurs through TRP channels (e.g., vanilloid class, TRPV4) to help refill ER Ca^2+^ stores while integral to the secondary “plateau” phase of the ∆[Ca^2+^]_i_ following G_q_PCR stimulation. The increase in [Ca^2+^]_i_ leads to production of nitric oxide (NO) and/or activation of small- and intermediate-conductance Ca^2+^-activated K^+^ (SK_Ca_ and IK_Ca_) channels. The production of NO is dependent on the conversion of l-arginine to l-citrulline in the presence of O_2_ via endothelial NO synthase (eNOS). Endothelial NOS contains an oxygenase domain to bind l-arginine, heme, Zn^2+^, and an essential cofactor tetrahydrobiopterin (BH_4_); a calmodulin (CaM) domain to bind Ca^2+^; and a reductase domain that binds to reducing agents nicotinamide adenine dinucleotide phosphate (NADPH), flavin mononucleotide (FMN), and flavin adenine dinucleotide (FAD). Activation of SK_Ca_ and IK_Ca_ channels generates hyperpolarization that is transmitted concomitantly along endothelial cells and to smooth muscle cells through myoendothelial gap junctions. Note that, regardless of endothelial Ca^2+^ mobilizing mechanism, endothelial [Ca^2+^]_i_ increases can regulate vascular tone ranging from normalization of vasoconstriction for basal tone (steady-state blood flow) to net decreases in vascular resistance (increased blood flow). Smooth muscle: Endothelium-derived hyperpolarization deactivates L-type voltage-gated Ca^2+^ channels (VGCCs) to prevent Ca^2+^ entry into smooth muscle cells (see broken lines with flat ends). Additionally, endothelial production of NO diffuses to smooth muscle and increases the activity of K^+^ channels such as the large-conductance Ca^2+^-activated K^+^ (BK_Ca_) channel via cGMP-dependent protein kinase (PKG) for hyperpolarization, another signaling input for deactivation of L-type VGCCs. Although not covered in detail in this review, TRP channels (TRPs; e.g., TRPM4) are also expressed in smooth muscle cells and play a general role for depolarization of V_m_, thereby activating L-type VGCCs for myogenic constriction. Note that, regardless of smooth muscle Ca^2+^ mobilizing mechanism, smooth muscle [Ca^2+^]_i_ increases can regulate vascular tone ranging from normalization of vasoconstriction for basal tone (steady-state blood flow) to net increases in vascular resistance (decreased blood flow). Myoendothelial feedback: Activation of α-adrenergic receptors (αARs) on smooth muscle by norepinephrine (NE) secreted by the SNS or the pharmacological α_1_R agonist phenylephrine (PE) results in IP_3_ production to elicit Ca^2+^ release through IP_3_Rs in the sarcoplasmic reticulum (SR) to evoke vascular contraction. When elevated in smooth muscle, IP_3_ and Ca^2+^ diffuse through myoendothelial gap junctions into the endothelium to activate SK_Ca_/IK_Ca_ channels and/or NO production, providing negative feedback to smooth muscle contraction (see broken lines indicating signaling from endothelium back to smooth muscle). The (−) and (+) symbols indicate V_m_ hyperpolarization and depolarization respectively while the respective color of lines corresponds to Ca^2+^ (red), IP_3_ (lime green), or Na^+^ (light blue) signaling.

**Figure 2 ijms-20-01380-f002:**
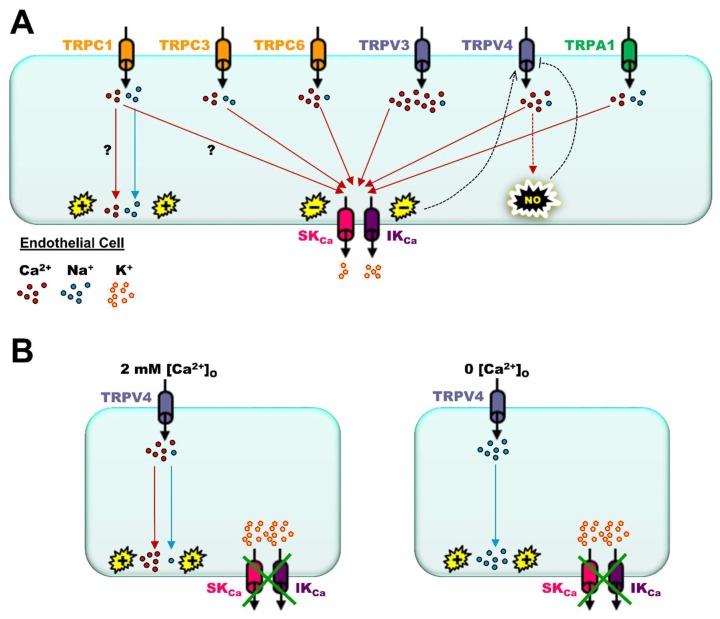
Functional contribution of major transient receptor potential (TRP) channel isoforms to activation of small- and intermediate-Ca^2+^-activated K^+^ (SK_Ca_ and IK_Ca_) channels and endothelium-derived hyperpolarization. (**A**) Permeability of Ca^2+^ to Na^+^ ions (P_Ca_/P_Na_) varies across the canonical (TRPC), vanilloid (TRPV), and ankyrin families of TRP channels. However, with the exception of TRPC1, endothelial TRP channels underlie significant influx of Ca^2+^ to activate SK_Ca_ and IK_Ca_ channels. As discussed in the text, note that these individual TRP isoforms can form tetrameric TRP channels via heteromeric combinations (e.g., TRPV4–TRPC1, TRPC3–TRPC4) in a physiological setting that may not be representative of determinations of homomeric channels expressed in heterologous culture systems. As it relates to the competition with Na^+^ influx, the “?” symbol indicates the unknown contribution of Ca^2+^ influx through TRPC1-containing channels for V_m_ depolarization vs. activation of SK_Ca_ and IK_Ca_ channels for V_m_ hyperpolarization as both are theoretically possible. Hyperpolarization via activation of SK_Ca_/IK_Ca_ channels may stimulate Ca^2+^ influx through TRPV4-containing channels for further activation of SK_Ca_ and IK_Ca_ channels and production of NO (“positive” feedback; broken black arrow). Although, nitric oxide (NO) may inhibit TRPV4-containing channels (“negative” feedback) via *S*-nitrosylation (broken black line with flat end). (**B**) In the absence of functional SK_Ca_ and IK_Ca_ channels, Ca^2+^ and Na^+^ influx (P_Ca_/P_Na_ ~6:1; extracellular Ca^2+^ ([Ca^2+^]_o_) present) through TRPV4-containing channels leads to endothelial depolarization (**left panel**). In the absence of [Ca^2+^]_o_, Na^+^ influx through TRPV4 channels is robust, and depolarization occurs regardless of SK_Ca_ and IK_Ca_ channel presence and function (**right panel**). Green crosses over SK_Ca_ and IK_Ca_ channels denote their functional absence in the plasma membrane (e.g., collecting lymphatic endothelium). For all panels, (−) and (+) symbols indicate V_m_ hyperpolarization and depolarization respectively while the respective color of lines corresponds to Ca^2+^ (red) or Na^+^ (light blue) signaling. All lines with arrow ends indicate positive signaling of respective ions to ultimately effect a change in V_m_ (hyperpolarization or depolarization) or production of NO.

**Figure 3 ijms-20-01380-f003:**
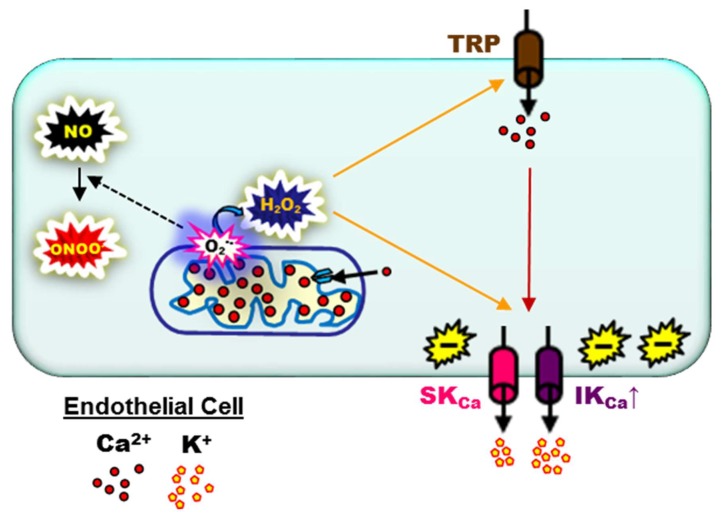
Working model for the upregulation of endothelium-derived hyperpolarization during aging and development of chronic pathology per enhanced oxidative signaling. Cardiovascular aging and the development of chronic disease is associated with a progressive increase in endothelial oxidative signaling. Mitochondrial respiration is a primary source of superoxide (O_2_^•−^) which inactivates NO to peroxynitrite (ONOO^−^) and, via superoxide dismutase, O_2_^•−^ is rapidly converted to H_2_O_2_. H_2_O_2_/OH^•^ activates SK_Ca_ and IK_Ca_ (primarily IK_Ca_) channels directly and/or indirectly (Ca^2+^ influx through TRP channels). Thus, SK_Ca_/IK_Ca_ channel function may “compensate” for decreased NO bioavailability to sustain local vasodilation. The (−) symbol indicates V_m_ hyperpolarization while the respective color of lines corresponds to O_2_^•−^ (black), H_2_O_2_ (orange), or Ca^2+^ (red) signaling.

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
