# Peer review of "Functional Interaction among KCa and TRP Channels for Cardiovascular Physiology: Modern Perspectives on Aging and Chronic Disease"

_ijms, 2019, doi:10.3390/ijms20061380_

Reviewer 1 Report

Summary:

Behringer et al, present a comprehensive review on the functional role of SKCa, IKCa and TRP channels in mediating vascular tone. They discuss the crosstalk between these channels and across the endothelium to underlying smooth muscle cells. Additionally, they discuss how these channels affect the membrane potential of the endothelium and regulate TRP channel function. The authors integrate a relevance to aging and chronic disease by summarizing data available for mostly rodent models. Overall, the authors clearly convey their excellent grasp of the topic, the manuscript is clearly written and the work is important to the calcium signaling, cardiovascular and chronic-disease studying communities. I have only some minor concerns which would improve the manuscript if addressed in a revised version.

Minor concerns:  

 1.     Formatting such as superscripts and subscripts are missing throughout. The entire manuscript should be carefully checked for word/font formatting issues.

 2.     Lines 44-45 – Endothelial cells also have Orai1 channels that can mediate store-operated calcium entry. This needs to be discussed in the same context as the activation of TRP channels. Smooth muscle cells also have Orai1. What is the role of STIM1 and Orai1 in the regulation of vascular tone?

 3.     Line 68-69 – Are TRP channels present only in endothelial cells? What about the vascular smooth muscle cells?

 4.     Figure 1 – The protein(s) involved in NO production should be added to the figure.

 5.     The word ‘interaction’ is being used in the discussion of intracellular calcium and membrane potential. A better word may be “crosstalk”, “relationship” or “association”. For the Int J Mol Sci, “interaction” may be confused with molecular interactions.

 6.     Lines 118-119 - why would CaM only use the N-lobe for Ca2+ sensing in the activation of SKCa/IKCa. This point requires some more detail.

 7.     Lines 128-129 - It would be useful to the reader if the specific connexins involved in forming the endothelial:smooth muscle gap junctons were listed.

 8.     Lines 141-143 - If the authors believe that the arrow representations in this part of the manuscript are providing important summary/information, I would suggest that they should incorporate this scheme into a new figure or figure panel.

 9.     Section 3 - The functional significance of the differences in K0.5 and Hill coefficients for SKCa and IKCa should be discussed.

 10.  Figure 2 – Integration of the possible heterooligomers discussed in the legend into the figure would enhance the figure and manuscript.

 11.  Lines 368-369 - Consider explicitly stating what oxidative-stress-related diseases you are referring to in this instance. There are myriad possibilities.

 12.  Figure 3 – Shows a hypothetical pathway for oxidative stress-induced modification of TRP channels. Can TRP channels be directly modified by NO – for example via S-nitrosylation?

 13.  Have there been any inheritable disease-associated mutations in IKCa or SKCa identified? Trp channels?  Connexins? These ion channels discussed in this manuscript, and some summary of the diseases (particularly if they are cardiovascular in nature) would enhance the manuscript.

 14.  A brief overview of how Ca2+ mediates smooth muscle cell contraction would be useful in this manuscript; prior to the discussion of dilation.

Author Response

The authors thank the reviewer for their thorough analysis of the manuscript towards overall improvement. Please see our responses and respective alterations to the manuscript below.

(1) In accord with our knowledge for journal instructions, we have ensured proper and consistent formatting throughout the manuscript.

(2) Yes, we thank the reviewer for the suggestion to provide more mechanistic resolution to the storage-operated (and in some cases, storage-independent) Ca2+ entry mechanism in vascular cells among other cell types. We have added a sentence accordingly (Lines 45-50) with suggestions for a recent primary study, two reviews, and a book chapter. Although a straightforward mechanism for explaining the physiology of Ca2+ entry, we prefer to keep discussion brief regarding the STIM-Orai mechanism as it is the least known in vascular cells (vs. TRPV4, etc.) with relatively few investigators at the helm in this regard. We look forward to seeing more primary work emerge from this area of investigation.

We have now clarified the role of Ca2+ entry in smooth muscle as well (Lines 101-106, 156-161, 311-315). Also, see our response to Comment #3.

(3) TRP channels are indeed expressed in both cell types. We have clarified this in Figure 1 and Figure 1 Legend (Lines 101-106, 311-315).

(4) We have added proteins and cofactors underlying the production of NO in Figure 1 with additional clarification in the legend (Lines 86-91).

(5) Where appropriate, we have removed ‘interaction’ from the manuscript regarding intracellular calcium and membrane potential and have substituted with “relationship” or “crosstalk” (Lines 37, 113, 163).

(6) This is a good point. During physiological conditions, it is likely that both the C-lobe and N-lobe play a critical role for KCa stabilization and activation in response to elevated cytosolic Ca2+. However, in patch-clamp measurements of Xenopus oocytes, it was found that the role of the N-lobe in particular appears indispensable for KCa function vs. the C-lobe. We have clarified this statement accordingly (Lines 132-137).

(7) We have specified the specific connexin isoforms that compose myoendothelial gap junctions (Lines 143-146).

(8) Figure 1 is intended to convey this information as an illustration. We have added final sentences to the “Endothelium” (Lines 93-96) and “Smooth muscle” (Lines 104-106) sections while referencing the Figure 1 Legend specifically in the text (Line 157).

(9) To properly establish clarity of functional interpretation of electrophysiological data for SKCa and IKCa channels, we have added more information regarding parameters of ion channel activity (e.g., probability of opening) and its dependence on various factors (e.g., overall affinity to Ca2+, cooperativity of channel subunits) (Lines 195-209, 211-217, 230-234). Thank you for prompting this expansion of information in order for the reader to better understand the functional significance of the channels based on complex patch clamp measurements and respective current-voltage plots.

(10) We specify our focus on TRPs integral to EDH while providing examples of common heterooligomers in the text (Lines 289, 292-293) as well as Figure 2 Legend (Lines 327-328).

(11) As this study was not focusing on a disease in particular, we have removed any discussion of “disease” here (Line 392). We elaborate on the myriad of possibilities involving enhanced oxidative signaling (e.g., diabetes, hypertension, etc.) in the current Section 6 (“Endothelial SKCa and IKCa during aging and chronic pathology”; Lines 484-516) with additional perspective in Figure 3 and Figure 3 Legend (Lines 518-525).

(12) This is a point we originally intended for Figure 2A. We have clarified this possibility in the Figure 2 Legend (Lines 329-330). We have indicated this possibility with a citation and a reference to Figure 2A in the text as well (Line 371).

 Finally, Figure 3 is indeed a working model and we have specified this in the title of the Figure 3 Legend (Line 518).

(13) We have added a new section 5 (“Role of familial mutations in KCa and TRP channels in the emergence of cardiovascular disease”) to discuss major discoveries regarding familial mutations and perspective for proceeding with central investigation in aging and chronic pathology (Lines 448-481).

(14) In accord with reviewer’s comments, we have endeavored to make the distinction of Ca2+ signaling in smooth muscle vs. endothelium (e.g., Figure 1, Figure 1 Legend, Lines 156-161; see responses to comments #2, #3, & #8) without deviating from the central focus of the manuscript significantly.

Reviewer 2 Report

This review by Behringer et al. summarizes a very complex topic of cardiovascular physiology in a very detailed and comprehensive way. The authors discuss the diverse interactions among KCa and TRP channels under various aspects of physiology and pathology. In this respect a recent review by Simonsen et al. (Acta Physiol. 219, 176-187 (2017)) is of interest here. It summarizes the synergistic interplay between TRPV4 and KCa3.1 contributing to endothelium-dependent vasodilation. Some of these aspects should be considered and discussed in the review by Behringer et al.

Minor point: Some of the references seem to be incomplete.

Author Response

(1) The authors thank the reviewer for their suggestion to include the citation of a literature review pertinent to the current manuscript. As a major component of discussion for the Simonsen et al review, we indeed discussed and cited the Wandall-Frostholm, C. et al study [Br J Pharmacol. 172(18), 4493-4505 (2015)]. Although, we have now directed readers to the Simonsen et al. for a thorough review on the interaction of pulmonary KCa and TRPV4 channels (Lines 546-547).

(2) We have ensured that the list of references cited are complete and accurate. Also, where reasonable, we have endeavored to further clarify the significance of individual studies throughout the manuscript while appropriately expanding the citation record within the scope of the manuscript (Lines 132-137, 211-217, 230-234, 310-314, 329-330, 448-481).

Reviewer 3 Report

Behringer et al present a review entitled "Functional interaction among KCa and TRP channels 3 for cardiovascular physiology: Modern perspectives on aging and chronic disease". This interesting reviews sumarizes the recent findings on TRP and calcium channles in the context of vascular physiology. The review then examines the roles of calcium and TRP channels in the context of vascular disease and vascular ageing. 

I feel that some recapitulating tables woudl be useful for the reader. Sugggestions : a table comparing the expression/behaviour of the principal TRP channels and KCa channels in normal and pathological conditions would be useful. Another table could summarize the main findings in vitro and in vivo in the context of each of the main channels.

A figure would be useful to illustrate the last sub-chapter "progressive recruitment of state-of the-art methods in the form of genetics, pharmacology, and engineering to examine classical components of genetics, structure, and function using cell/tissue/animal models of aging". the authors could precise the vectors used (viruses, nanothechnology etc.)

Minor:

lines 543- : please remove "although" from "Although, H2S generated by cystathionine γ-lyase in particular may activate 537 endothelial SKCa and IKCa channels for vasodilation of mouse mesenteric arteries". In addition, although is repeated in the next sentence.

Please clarify "Due to increasing legalization and social acceptance year-over-year since 2007 for recreational and medicinal use of cannabis among American populations, perhaps investigation of cannabinoids in particular is among the highest of priorities for current cardiovascular research ". it is unclear and should be cut in two.

Lines 571-572. Please rephrase"...endothelium-derived hyperpolarization (EDH); entails Gq-protein coupled receptor stimulation in intracellular Ca2+ release from the endoplasmic reticulum and Ca2+ influx"

Author Response

The authors thank the reviewer for their suggestion to include additional Tables and a Figure while calling attention to some ambiguous statements in the manuscript. Please see the responses of the authors below.

(1) This suggestion by the reviewer to summarize essentially the entire manuscript into additional Tables is well-appreciated. Although, we prefer to keep the illustrations as simple as possible and indeed, Figures 1 & 2 present the interaction of TRP channels and KCa channels during normal conditions and then we focus on a potential pathological mechanism as Figure 3. For the second table suggestion, the majority of studies performed were actually ex vivo as the best compromise for physiological accuracy and the demands of understanding ion channel mechanisms via scientific reductionism.

(2) This suggestion to add a Figure on future directions is also helpful. We do not feel that there is sufficient information yet in order to prepare a comprehensive Figure to our satisfaction. Although, we have now added references to indicate some directions to think on for the future (Lines 625-626). As to not deviate outside of the scope of the current manuscript, perhaps it would be best for us to prepare another “frontiers” type review manuscript of this type to convey in detail for potential readers.

(3) Thank you. We have eliminated “Although” from this sentence (Line 583).

(4) We have divided this sentence into two sentences while clarifying the grammar (Lines 590-593).

(5) We have divided this sentence into two sentences as well (Lines 615-619).